# Reversible Thiol Oxidation Inhibits the Mitochondrial ATP Synthase in *Xenopus laevis* Oocytes

**DOI:** 10.3390/antiox9030215

**Published:** 2020-03-05

**Authors:** James Cobley, Anna Noble, Rachel Bessell, Matthew Guille, Holger Husi

**Affiliations:** 1Centre for Health Sciences, University of the Highlands and Islands, Inverness IV2 3JH, UK; rbessell@outlook.com (R.B.); holger.husi@uhi.ac.uk (H.H.); 2School of Biological Sciences, European Xenopus Resource Centre, University of Portsmouth, King Henry Building, Portsmouth PO1 2DY, UK; anna.noble@port.ac.uk (A.N.); matthew.guille@port.ac.uk (M.G.)

**Keywords:** mitochondria, thiol, redox signaling, ATP synthase, oocyte, *Xenopus laevis*, click chemistry

## Abstract

Oocytes are postulated to repress the proton pumps (e.g., complex IV) and ATP synthase to safeguard mitochondrial DNA homoplasmy by curtailing superoxide production. Whether the ATP synthase is inhibited is, however, unknown. Here we show that: oligomycin sensitive ATP synthase activity is significantly greater (~170 vs. 20 nmol/min^−1^/mg^−1^) in testes compared to oocytes in *Xenopus laevis* (*X. laevis*). Since ATP synthase activity is redox regulated, we explored a regulatory role for reversible thiol oxidation. If a protein thiol inhibits the ATP synthase, then constituent subunits must be reversibly oxidised. Catalyst-free *trans*-cyclooctene 6-methyltetrazine (TCO-Tz) immunocapture coupled to redox affinity blotting reveals several subunits in F_1_ (e.g., ATP-α-F_1_) and F_o_ (e.g., subunit c) are reversibly oxidised. Catalyst-free TCO-Tz Click PEGylation reveals significant (~60%) reversible ATP-α-F_1_ oxidation at two evolutionary conserved cysteine residues (C^244^ and C^294^) in oocytes. TCO-Tz Click PEGylation reveals ~20% of the total thiols in the ATP synthase are substantially oxidised. Chemically reversing thiol oxidation significantly increased oligomycin sensitive ATP synthase activity from ~12 to 100 nmol/min^−1^/mg^−1^ in oocytes. We conclude that reversible thiol oxidation inhibits the mitochondrial ATP synthase in *X. laevis* oocytes.

## 1. Introduction

Human sperm rely on oxidative phosphorylation (OXPHOS) to swim 10^3^ times their own length to fertilise an oocyte [1,2]. Paternal mitochondrial DNA (mtDNA) is purged and/or heavily diluted after fertilisation to ensure maternal inheritance dominates in the embryo [3,4]. Maternal inheritance avoids deleterious mtDNA heteroplasmy because OXPHOS sensitises sperm to oxidative DNA damage [5]. Oxidative DNA damage can occur when thermodynamically and kinetically competent reduced electron donors (e.g., prosthetic semiquinone radicals) catalyse the univalent reduction of ground state molecular dioxygen (O_2_) to superoxide [6,7]. Superoxide anion and its dismutation product hydrogen peroxide (H_2_O_2_) are chemically unable to oxidise DNA directly [8]. H_2_O_2_ can, however, react with DNA bound iron and copper ions to produce hydroxyl radical (OH•) [9,10,11]. In turn, OH• can damage pyrimidine and purine bases at a diffusion controlled rate (i.e., *k* ~ 10^9^ M^−1^ s^−1^) via addition, oxidation, and abstraction reactions [12,13,14]. If OXPHOS imperils mtDNA homoplasmy, then oocyte mitochondria may repress it to curtail superoxide production. 

Allen [15] posits that: oocytes safeguard mtDNA homoplasmy by repressing OXPHOS to curtail superoxide production. In support, OXPHOS is repressed in oocytes compared to sperm in diverse phyla from jellyfish to mice [16,17,18,19,20]. Repressed OXPHOS is associated with lower mitochondrial free radical levels in oocytes compared to sperm [16,17]. To curtail superoxide production by repressing OXPHOS without sacrificing oocyte viability, dual inhibition of the proton pumps (i.e., complex I, III and IV) and F_1_-F_o_ ATP synthase may be required. If the proton pumps are active and the F_1_-F_o_ ATP synthase is inactive, then a large electrochemical proton motive force (Δ*p*) could substantially enhance superoxide production (e.g., by complex I catalysed reverse electron transfer [21]). Reciprocally, if the proton pumps are inactive and F_1_-F_o_ ATP synthase is active, then it may curtail complex I and III catalysed superoxide production, but the synthase may compromise oocyte viability by hydrolysing ATP to maintain Δ*p* [22]. Whether the proton pumps and F_1_-F_o_ ATP synthase are inhibited in oocytes is, however, unknown. Unravelling if and how the F_1_-F_o_ ATP synthase is inhibited would advance current understanding of reproductive biology. 

Extant data imply the F_1_-F_o_ ATP synthase is inhibited in *Xenopus laevis* (*X. laevis*) oocytes. In support, the F_1_-F_o_ ATP synthase inhibitor oligomycin fails to deplete [ATP] in rapidly proliferating *X. laevis* blastulae [23,24]. Their oligomycin insensitivity may be explained by pre-existing inhibition by reversible thiol oxidation. Indeed, we observed substantial reversible thiol oxidation of the F_1_ alpha subunit (ATP-α-F_1_) in *X. laevis* oocytes [25]. Informed by seminal work in chloroplasts and somatic mitochondria [26,27,28,29,30,31,32,33], we infer that the F_1_-F_o_ ATP synthase is inhibited by reversible thiol oxidation; which can tune protein function by modifying activity, subcellular locale, and/or vicinal interactome (reviewed in [34,35,36,37,38]). Since Yagi and Hatefi [26] first reported that reversible thiol oxidation inhibits the F_1_-F_o_ ATP synthase in 1984, subsequent studies [29,32,33] have shown that it regulates OXPHOS, superoxide production, and the mitochondrial permeability transition pore (reviewed in [31,39,40,41]). For example, Wang and colleagues [29] found that a disulfide bond between the ATP-α-F_1_ and gamma (ATP-γ-F_1_) subunits impaired OXPHOS in dyssynchronous heart failure. 

No study has investigated whether reversible thiol oxidation inhibits the F_1_-F_o_ ATP synthase in oocytes. To advance current understanding, we determined whether: (1) F_1_-F_o_ ATP synthase activity is impaired in the female germline compared to the testes (i.e., a somatic tissue responsible for producing the male germline); (2) the F_1_-F_o_ ATP synthase is assembled; (3) F_1_-F_o_ ATP synthase subunits are reversibly oxidised; and (4) F_1_-F_o_ ATP synthase activity is redox regulated in *X. laevis* oocytes. *X. laevis* oocytes are ideal because they are a tractable developmental model [42,43,44], insensitive to oligomycin [45,46], and key thiols are conserved [25]. 

## 2. Materials and Methods

### 2.1. Materials and Reagents

A list of the materials and reagents used is provided (see Appendix A). 

### 2.2. Xenopus laevis

In-house bred *X. laevis* were maintained at the European *Xenopus* Resource Centre (EXRC) at 18 °C and fed daily on trout pellets [47]. Following ethical approval (#OLETHSHE1500), *X. laevis* oocytes were harvested, defolliculated with collagenase, and stored at −80 °C for biochemical analysis. In line with the ARRIVE guidelines [48], biological variability was accounted for by obtaining samples from three different adult females. 

### 2.3. F_1_-F_o_ ATP Synthase Assay

Mitochondria were isolated by differential centrifugation wherein oocytes (*n* = 10) were lysed in STE buffer (250 mM sucrose, 200 mM Tris-HCL, 2 mM EDTA, pH 7.2) supplemented with a protease inhibitor tablet, 1% fatty acid free BSA and 100 mM N-ethylmaleimide (NEM) to block reduced thiols for 10 min on ice. Lysates were centrifuged at 700× *g* for 10 min at 4 °C, before the supernatant was centrifuged at 7000× *g* for 10 min at 4 °C. After discarding the supernatant, the mitochondrial pellet was resuspended in STE with fresh 10 mM 1-4-Dithiothreitol (DTT) or without (control) for 30 min on ice. Mitochondria were pelleted and washed (3 × 1 min in BSA free STE) to remove excess DTT, before being treated with 50 µg/mL alamethicin to permeabilise the inner membrane to ATP [49], 1 µM diphenyleneiodonium to prevent complex I oxidising NADH by inhibiting the prosthetic flavin mononucleotide group, and 300 nM antimycin A to block complex III. In the reduced group, TCEP (2 mM) was used to maintain a reducing environment (e.g., prevent vicinal dithiols reforming disulfide bonds after reduction). TCEP is preferable to DTT for maintaining a reducing environment because DTT can autoxidise to produce superoxide in the presence of transition metals [12]. 

F_1_-F_o_ ATP synthase activity was assessed by monitoring ATP hydrolysis in the presence of a glycolytic pyruvate kinase (PK), lactate dehydrogenase (LDH), and phosphoenolpyruvate (PEP) system to regenerate ATP. ATP hydrolysis was followed as the decrease in NADH absorbance at 340 nm (extinction coefficient: 6.22 Mm^−1^ cm^−1^) using a plate reader. Mitochondria were analysed in duplicate in a reaction buffer containing (400 µM NADH, 1 mM PEP, 20 U/mL LDH, 15 U/mL PK, and 2.5 mM ATP in 200 mM Tris, 2 mM MgCl_2_, 200 µM EDTA, pH 8.0). ATP hydrolysis was followed for 2 min without mitochondria to account for spontaneous ATP hydrolysis. Mitochondria were added and NADH absorbance was monitored for every 15 s 10 min. All wells were then spiked with 6 µM oligomycin and NADH absorbance was followed for 5 min to determine specific F_1_-F_o_ ATP synthase activity. After determining protein content using a Bradford assay and subtracting oligomycin insensitive absorbance, F_1_-F_o_ ATP synthase activity (nmol/min^−1^/mg^−1^) was calculated using the following equation: (Δ 340/min × 1000)/[(extinction coefficient × sample volume) × protein concentration)] [50].

### 2.4. Native Blotting

Mitochondrial membranes were solubilised with 5 µL of 20% dodecyl lauryl maltoside (DDM) in PBS. After centrifuging samples at 16,250× *g* for 12 min at 4 °C, 0.1% Ponceau S was added to the supernatant to establish a dye front [51]. High resolution clear native polyacrylamide gel electrophoresis (Hr-CNPAGE) was performed as described by Wittig and colleagues [51] using a cathode (50 mM Tricine, 7.5 mM Imidazole, 0.02% DDM, and 0.05% sodium deoxycholate, pH ~7) and anode buffer (25 mM Imidazole, pH ~7). Electrophoresis was performed at 4 °C for 90 min to prevent band broadening and to ensure sufficient protein transfer [52]. Gels were transferred onto a low autofluorescence 0.45 µM polyvinylidene fluoride (PVDF) membrane at 100 V for 60 min in transfer buffer (50 mM Tricine, 7.5 mM Imidazole, pH 7) at 4 °C. Membranes were blocked with 5% non-fat dry milk (NFDM) in PBS for at least 60 min. Primary/secondary antibody incubations and fluorescent detection are described below (see Redox Mobility Shift Assay). 

### 2.5. In-Gel ATP Hydrolysis Assay

Mitochondrial membranes were solubilised with DDM and Hr-CNPAGE was performed as described above. After alkylating reduced thiols with NEM (100 mM), reversibly oxidised thiols were treated with (reduced) or without (control) 3.5 mM TCEP for 15 min. Oligomycin controls (500 nM) were loaded in parallel. An equal amount of protein (5 µg) was loaded onto a precast 4–15% gradient gel. Native gels were incubated with buffer (35 mM Tris, 270 mM Glycine, pH 8.3) for 2 h, before being incubated with assay buffer (35 mM Tris, 270 mM Glycine, 14 mM MgSO_4_, 0.2% Pb(NO_3_)_2_, 2.8mM ATP, pH 8.3) for 2 h at room temperature. The reaction was stopped with 50% methanol for 30 min before membranes were scanned. Images were inverted to visualise the white bands and densitometry was performed using VisionWorks^TM^ software (Analytik Jena, Germany).

### 2.6. Catalyst-Free TCO-Tz Immunocapture Coupled to Redox Affinity Blotting

Aliquots (5 µL) of anti-ATP-α-F_1_ primary antibody were incubated with 10 mM *trans*-cyclooctene polyethylene glycol 4 (PEG) N-hydroxysuccinimide (TCO-PEG4-NHS) for 30 min at 4 °C to label primary amines. To prevent unwanted labelling of primary amines in the sample, excess TCO-PEG4-NHS was quenched by adding 5 mM Tris for 15 min. A 45 min total reaction time should ensure NHS hydrolysis outcompetes the carbodiimide reaction thereby preventing adventitious sample labelling. Mitochondrial membranes were prepared as described below (see Catalyst-free TCO-Tz Click PEGylation), except TCO-PEG3-maleimide (TCO-PEG3-NEM) was substituted for 5 mM Biotin-dPEG^®^3-maleimide (Sigma, UK, Biotin-dPEG^®^3-MAL). Biotin-dPEG^®^3-MAL labelled mitochondrial membranes were incubated with TCO-PEG4-NHS labelled primary antibody for 30 min on ice, before being placed in a spin cup containing 85 µL 6-methyltetrazine substituted agarose beads to initiate the catalyst-free Inverse Electron Demand Diels Alder (IEDDA) Click reaction for 90 min on ice. Clicked samples were washed in PBS supplemented with 0.05% DDM for 1 min before being centrifuged for 1 min at 1000× *g* (washing was repeated five times). Using a spin cup increases purity by removing contaminants with high stringency. Western blotting was performed as described below (see Redox Mobility Shift Assay), expect Streptavidin Alexa Fluor™ 647 (ThermoFisher, UK, 1:500 in TBST for 60 min at room temperature) was used to detect reversibly oxidised subunits on an Analytik Jena (Analytik Jena, Germany) scanner using the appropriate filters (excitation: 600–645 nm; emission: 607–682 nm). 

### 2.7. Catalyst-Free TCO-Tz Click PEGylation 

We amended whole-cell TCO-Tz Click PEGylation for mitochondria [25]. Oocytes (*n* = 10) were lysed in STE buffer supplemented with a protease inhibitor tablet, 1% fatty acid free BSA and 100 mM NEM) to block reduced thiols for 10 min on ice. Lysates were centrifuged at 700× *g* for 10 min at 4 °C, before the supernatant was centrifuged at 7000× *g* for 10 min at 4 °C. Mitochondrial pellets were resuspended in STE with 5 mM TCEP for 30 min on ice. After washing to remove excess TCEP, mitochondria were resuspended in 5 mM TCO-PEG3-NEM to label newly reduced thiols [36]. Alamethicin (50 µg/mL) was added to ensure TCO-PEG3-NEM could permeate the inner mitochondrial membrane. Mitochondria were lysed in PBS (pH 7.3) supplemented with 1.5% (*v*/*v*) 20% DDM. After removing insoluble material by centrifugation (14,000× *g* for 5 min at 4 °C), the supernatant was incubated with 5 mM 6-methyltetrazine 5 kDa PEG (Tz-PEG5) for 90 min on ice. The catalyst-free IEDDA Click reaction was terminated by adding Laemmli buffer (4% SDS, 20% Glycerol, 0.004% Bromophenol blue and 125 mM Tris HCl, pH 6.8) supplemented with 100 mM DTT before samples were denatured at 80 °C for 5 min. 

### 2.8. Redox Mobility Shift Assay

The Redox Mobility Shift Assay and analysis were performed as described in [25]. After following a standard Western blot protocol [53,54], PVDF membranes were incubated with anti-ATP-α-F_1_ primary antibody (1 µg/mL in 3% NFDM TBST) overnight. Washed and incubated with a preabsorbed Alexa Fluor^®^750 secondary antibody (Abcam, UK, 1:2000 in 3% NDFM TBST). Membranes were imaged (excitation: 678–748 nm; emission: 767–807 nm) on an Analytik Jena scanner (Germany). 

### 2.9. Statistical Analysis

Oligomycin sensitive F_1_-F_o_ ATP synthase activity data were analysed using independent Student’s t-tests with alpha ≤ 0.05. TCO-Tz Click PEGylation data were analysed by paired Student’s t-tests with alpha ≤ 0.05. Statistical analysis was performed on GraphPad Prism (GraphPad Software, USA). Data are presented as Mean and standard deviation (±).

## 3. Results

### 3.1. F_1_-F_o_ ATP Synthase Activity is Significantly Greater in Testes Compared to Oocytes 

Allen [15] proposes that: oocytes repress OXPHOS to safeguard mtDNA homoplasmy by curtailing superoxide production. If so, F_1_-F_o_ ATP synthase activity should be greater in the soma compared to the female germline. Sperm, a major cell type in testes, should sacrifice mtDNA homoplasmy by practicing OXPHOS [15]. Isolating sperm without contaminating somatic tissue (i.e., sertoli cells) is problematic; a situation abetted by a five orders of magnitude difference in mitochondrial number between sperm (10^2^) and oocytes (10^7^) in *X. laevis* [55]. Accordingly, we assessed mitochondrial F_1_-F_o_ ATP synthase activity in oocytes compared to testes. We measured ATP hydrolysis using a glycolytic ATP regenerating system in isolated mitochondria treated with alamethicin [49] (Figure 1A). Using alamethicin to render the inner mitochondrial membrane permeable to ATP placed the rate-limiting step on the F_1_-F_o_ ATP synthase by eliminating the influence of the proton pumps and ATP/ADP carrier [22]. Oligomycin sensitive F_1_-F_o_ ATP synthase activity is significantly (*p* ≤ 0.0001) lower in oocytes compared to testes (oocyte: 18.81 ± 15.38; testes: 177.1 ± 37.82 nmol/min^−1^/mg^−1^; Figure 1B). 

### 3.2. The F_1_-F_o_ ATP Synthase is Assembled in Oocytes

Sieber and colleagues [45] identified a Coomassie stained band on a native gel that may correspond to assembled F_1_-F_o_ ATP synthase in *X. laevis* oocytes. To immunologically confirm F_1_-F_o_ ATP synthase assembly, we performed a native blot against the ATP-α-F_1_ subunit [51,52]. Native blotting reveals the F_1_-F_o_ ATP synthase is assembled in *X. laevis* oocytes (Figure 1C). F_1_-F_o_ ATP synthase disassembly is, therefore, unlikely to explain low oligomycin sensitive ATP hydrolysis in oocytes. Low activity is also unlikely to be attributable to low abundance because ATP-β-F_1_ content is estimated to be 7.1 µM in *X. laevis* oocytes [56]. F_1_-F_o_ ATP synthase assembly and abundance implies latent enzyme capacity that may be realised by reversing inhibitory thiol oxidation. 

### 3.3. Several F_1_-F_o_ ATP Synthase Subunits are Reversibly Oxidised 

If reversible thiol oxidation is inhibitory, then F_1_-F_o_ ATP synthase subunits must be reversibly oxidised. Annotated genome data reveals the F_1_-F_o_ ATP synthase contains 18 thiols in *X. laevis* [42] (see Table 1). After excluding mitochondrial leader sequences, the F_1_-F_o_ ATP synthase likely contains 10–11 thiols depending on whether the L or S chromosome copy of ATP-y-F_1_ is expressed. Available structures suggest cysteine residues in ATP-α-F_1_ (C^244^, C^294^), ATP-γ-F_1_ (C^100^, C^173^), OSCP (C^139^), and subunit c (C^149^) are likely fully and/or partially solvent exposed in catalytic state 3A [57] (Figure 2). Assuming a similar structure in *X. laevis*, C^104^ in subunit b may also be solvent exposed. To determine whether F_1_-F_o_ ATP synthase subunits are reversibly oxidised, we used a catalyst-free TCO-Tz immunocapture approach coupled to redox affinity blotting (Figure 3A). Since the ATP-α-F_1_ antibody recognises the assembled complex, we used a heterobifunctional TCO-PEG4-NHS linker to form a stable carbodiimide bond with primary amines in the ATP-α-F_1_ antibody. After labelling reversibly oxidised samples with Biotin-dPEG^®^3-MAL, samples were incubated with TCO-PEG4-NHS labelled ATP-α-F_1_ antibody before 6-methyltetrazine substituted agarose was used to capture the synthase. Reversibly oxidised F_1_-F_o_ ATP synthase subunits were detected by Western blot using a streptavidin conjugated fluorophore [36].

Catalyst-free TCO-Tz immunocapture coupled to redox affinity blotting reveals discrete bands at ~100, 50, 37, 30, 20–25, and ≤ 10 kDa. Additionally, a distorted band at ~10 kDa was observed, which may reflect how DDM and SDS interact with hydrophobic proteins. Based on a theoretical profile constructed from Table 1 (Figure 3B), observed bands likely correspond to ATP-α-F_1_, ATP-γ-F_1_, OSCP, subunit b and g (Figure 3C). For OSCP and subunit g, the observed band reflects reversible C^139^ and C^96^ oxidation, respectively, because they contain a single thiol. After excluding mitochondrial leader sequences, the distorted band is likely an F_o_ subunit (e.g., subunit c and/or C3). The unassigned ~100 kDa band may represent an interacting protein, hydrophobic aggregate (likely heat induced), and/or crosslinked subunits. Proteomic profiling of the captured complex will be reported elsewhere. F_1_-F_o_ ATP synthase subunits are reversibly oxidised in oocytes. 

### 3.4. Reversible ATP-α-F_1_ Oxidation is Significant in Oocytes 

After demonstrating that F_1_-F_o_ ATP synthase subunits are reversibly oxidised, we immunologically verified an observed band. To do so, we assessed whether ATP-α-F_1_ is reversibly oxidised at two evolutionary conserved cysteine residues (C^244^ and C^294^) using catalyst-free TCO-Tz Click PEGylation [25]. TCO-Tz Click PEGylation exploits catalyst-free IEDDA chemistry [58,59,60] to selectively conjugate a low molecular weight (5 kDa) PEG moiety to reversibly oxidised thiols. Selectively conjugating PEG imparts an electrophoretic mobility shift to render reversibly oxidised thiols detectable as mass shifted bands by Western blotting [25,61,62] (Figure 4A). Consistent with our previous work [25], TCO-Tz Click PEGylation reveals that: 62.9 ± 3.0% of total ATP-α-F_1_ is reversibly oxidised in oocytes (Figure 4B). Percent reversibly oxidised ATP-α-F_1_ is significantly (*p* = 0.0007) greater than the amount of reduced ATP-α-F_1_ (Figure 4C). No significant difference (*p* = 0.9611) in the contribution of the 5 (49.8 ± 10.4%) and 10 (50.2 ± 10.4%) kDa bands to total percent reversibly oxidised ATP-α-F_1_ was observed (Figure 4D). TCO-Tz Click PEGylation reveals ~20% (2 out of 10 or 11) of the total thiols in the F_1_-F_o_ ATP synthase are substantially oxidised. 

### 3.5. Reversible Thiol Oxidation Inhibits the F_1_-F_o_ ATP Synthase 

Having established several subunits are reversibly oxidised, we explored whether F_1_-F_o_ ATP synthase activity is redox regulated. If reversible thiol oxidation inhibits the F_1_-F_o_ ATP synthase, then chemically reducing oxidised thiols using TCEP should increase F_1_-F_o_ ATP synthase activity in oocytes. TCEP significantly (*p* = 0.0007) increases oligomycin sensitive F_1_-F_o_ ATP synthase activity in oocytes (TCEP: 102.9 ± 44.97; Control: 12.90 ± 7.3 nmol/min^−1^/mg^−1^; Figure 5A). Alkylating reduced thiols with 5 mM NEM means ex vivo thiol oxidation is unlikely to constrain F_1_-F_o_ ATP synthase activity in the control. Previous work excluded the possibility that: 5 mM NEM inhibits F_1_-F_o_ ATP synthase activity [29]. To be sure, we assessed F_1_-F_o_ ATP synthase catalysed ATP hydrolysis using Hr-CNPAGE [51]. In this assay, F_1_-F_o_ ATP synthase catalysed ATP hydrolysis is detected as white lead phosphate precipitates (Figure 5B). TCEP significantly increased (*p* = 0.0067) F_1_-F_o_ ATP synthase mediated ATP hydrolysis in oocytes (TCEP: 111.8 ± 10.0; Control: 75.0 ± 7.2; Figure 5C). 

## 4. Discussion

We advance knowledge of reproductive biology by showing, for the first time, that the F_1_-F_o_ ATP synthase is inhibited in *X. laevis* oocytes. Repressed F_1_-F_o_ ATP synthase activity could impair OXPHOS by uncoupling Δ*p* from ATP synthesis, but may enhance oocyte viability by constraining ATP hydrolysis. In considering Allen’s theory [15], low F_1_-F_o_ ATP synthase activity would curtail superoxide production to safeguard mtDNA homoplasmy provided the proton pumps were inhibited. In support, cytochrome c oxidase (i.e., complex IV) activity is repressed and H_2_O_2_ levels are low in *X. laevis* oocytes [23,46]. Complete proton pump inhibition is, however, unlikely because oocyte mitochondria sustain an Δ*p* and still produce some superoxide [23,63]. Protein binding to shield mtDNA and/or selectively curtailing superoxide production at single site (e.g., complex I) may, therefore, be required to safeguard mtDNA homoplasmy. Follow-up studies are required to determine proton pump activity and superoxide production in *X. laevis* oocytes using state-of-the-art tools (e.g., MitoNeoD [64]). 

Existing literature has firmly established that F_1_-F_o_ ATP synthase activity is redox regulated in somatic mitochondria [26,27,28,29,30,31,32,33]. Current understanding is, however, restricted to isolated organelles and/or disease models. Whether redox regulation plays a physiological role is, therefore, unclear. We make a major novel contribution by showing that: chemically reversing protein thiol oxidation significantly increases F_1_-F_o_ ATP synthase activity in *X. laevis* oocytes. Our result defines a novel physiological role for mitochondrial reversible thiol oxidation in reproductive biology. Using a redox switch to inactivate the synthase during oogenesis would only imperil mtDNA homoplasmy if any damage sustained was unrepaired. Importantly, cells can transduce redox signals without sustaining oxidative macromolecule damage [65]. The ability of a redox switch to hold the F_1_-F_o_ ATP synthase inactive informs several hypotheses. For example, a protein thiol could regulate the metabolic switch from a reliance on fermenting glucose to OXPHOS in the developing *X. laevis* retina [66]. If a such a developmental Warburg phenotype is redox regulated, it may help rationalise how mitochondrial oxidative stress rewires metabolism in cancer [67]. 

Unravelling how reversible thiol oxidation inhibits the F_1_-F_o_ ATP synthase in oocytes relies on using redox proteomics and site-directed mutagenesis to identify the redox switch(es) (i.e., subunit (s) and site (s)) [68,69,70]. Disambiguating reversible modification type is also important because S-glutathionylation and disulfide bonds may inhibit the enzyme by different mechanisms [39]. Biological precedent exists: ATP-α-F_1_ S-glutathionylation seems to electrostatically repel nucleotide binding by introducing a bulky negative charge whereas intermolecular disulfide bonds between ATP-α-F_1_ and ATP-γ-F_1_ may impair conformational flexibility [30]. Reversible thiol oxidation in F_o_ is likely to inhibit catalysis by disrupting the ability to bind protons and/or rotate the c-ring [31,39,40,41,71]. For example, reversible oxidation of subunit c at C^84^ could impede proton transport owing to its proximity to glutamate 58 [72,73]. Reversible thiol oxidation may also protect the F_1_-F_o_ ATP synthase from irreversible inactivation secondary to sulfinic and sulfonic acid formation [25]. If reversible thiol oxidation impacts inhibitor binding and/or action, then it may lead to oligomycin sensitive enzyme activity being underreported. Given reversible thiol oxidation can activate several enzymes, it is unwise to assume reversible thiol oxidation is always inhibitory. Intriguingly, a redox code may exist wherein the biological outcome varies according to the number of thiols and subunits modified, occupancy (percent oxidation), and modification type [65]. 

From a structural perspective, S-glutathionylation could lock the F_1_-F_o_ ATP synthase in a monomeric state by impeding the formation of dimers, especially if they involve intermolecular disulfide bonds [31]. Alternatively, a negative charge may electrostatically repel dimer interfaces. Thiols in subunit e and g also regulate the stability and formation of oligomers, which dictate inner mitochondrial membrane topology [74,75,76]. For example, F_1_-F_o_ ATP synthase dimers shape cristae to create an efficient proton sink for ATP synthesis [77]. Perhaps, F_o_ redox state underlies the lack of mature cristae in oocytes [19]. Teixeira and colleagues [78] found that OXPHOS is dispensable for germline stem cell differentiation but F_1_-F_o_ ATP synthase dimers are essential because they are required for cristae maturation. The results of their study and the present work raise the possibility of at least two redox switches: (1) to prohibit dimers; and (2) to inhibit catalysis. Two redox switches would endow mitochondria with the capacity to differentiate cristae without a catalytically active F_1_-F_o_ ATP synthase. That is, to uncouple cristae differentiation from OXPHOS [78]. 

## 5. Conclusions

We conclude that: (1) F_1_-F_o_ ATP synthase activity is significantly greater in testes compared to oocytes; (2) F_1_-F_o_ ATP synthase subunits are reversibly oxidised in oocytes; (3) reversible ATP-α-F_1_ oxidation at evolutionary conserved cysteine residues (C^244^ and C^294^) is substantial (~60% of total ATP-α-F_1_) in oocytes; and (4) chemically reversing thiol oxidation significantly increases F_1_-F_o_ ATP synthase activity. Reversible thiol oxidation, therefore, inhibits the mitochondrial ATP synthase in *X. laevis* oocytes. 

## Figures and Tables

**Figure 1 antioxidants-09-00215-f001:**
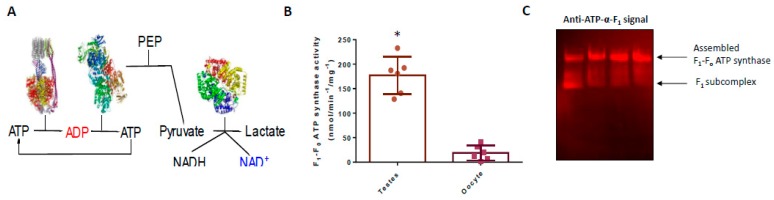
Oligomycin sensitive F_1_-F_o_ ATP synthase activity is significantly greater in testes compared to oocytes. (**A**) The F_1_-F_o_ ATP synthase hydrolysis ATP to ADP. Pyruvate kinase regenerates ATP by using phosphoenolpyruvate (PEP) to phosphorylate ADP to ATP. Lactate dehydrogenase reduces pyruvate to lactate using NADH derived electrons. F_1_-F_o_ ATP synthase activity is followed by monitoring the loss of NADH absorbance at 340 nm. (**B**). Oligomycin sensitive F_1_-F_o_ ATP synthase activity is higher significantly (*p* ≤ 0.0001) in testes (*n* = 6) compared to oocytes (*n* = 6) in *X. laevis*. Statistical significance is indicated by an asterix as assessed by an independent Student’s t-test. (**C**). Native ATP-α-F_1_ blot image showing the F_1_-F_o_ ATP synthase is fully assembled in *X. laevis* oocytes (*n* = 4). A minor fraction is present as an F_1_ subcomplex. Each *n* is the weighted mean of 10 oocytes.

**Figure 2 antioxidants-09-00215-f002:**
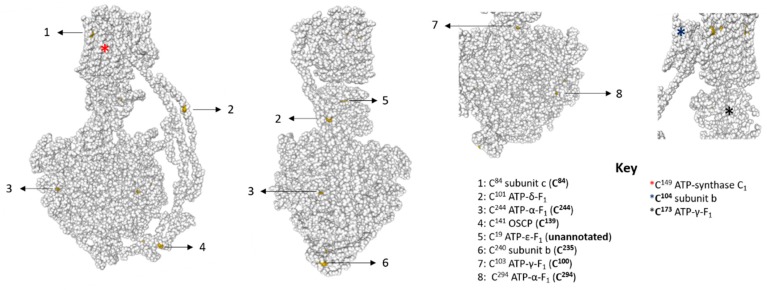
Cysteine residues in bovine F_o_-F_1_ ATP synthase in catalytic state 3A. Numbered cysteine residues are highlighted in yellow. The key lists the cysteine residue by amino acid number for the bovine enzyme with the equivalent *X. laevis* residue in brackets in bold. A coloured asterix denotes the predicted position of additional cysteine residues in *X. laevis.* Additional cysteine residues are highlighted in bold in the key if they are likely to be solvent exposed. No structural data is available for subunit g and C3.

**Figure 3 antioxidants-09-00215-f003:**
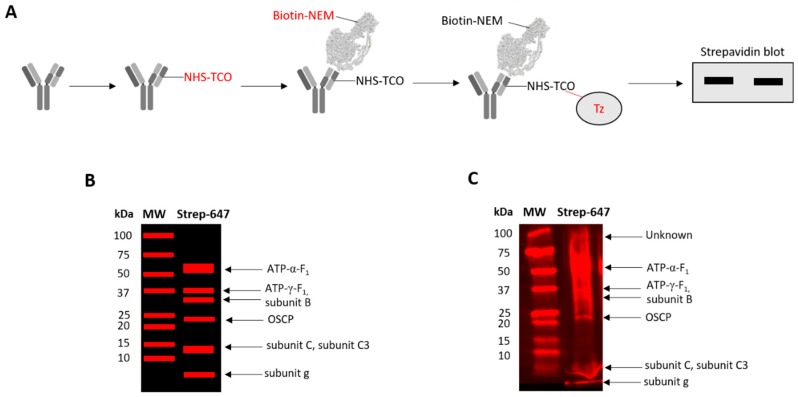
Several F_1_-F_o_ ATP synthase subunits are reversibly oxidised. (**A**). Catalyst-free trans-cyclooctene-6methyltetrazine (TCO-Tz) immunocapture coupled to redox affinity blotting workflow. From left to right: Primary amines in the ATP-α-F_1_ antibody are labelled with a heterobifunctional NHS-PEG4-TCO linker. After excess NHS is quenched with Tris (not shown), the labelled antibody is incubated with Biotin functionalised maleimide labelled reversibly oxidised thiols in mitochondrial membranes to capture the F_1_-F_o_ ATP synthase. Agarose beads substituted with 6-methyltetrazine are then used to selectively capture the antibody-synthase complex. After washing away contaminants with a spin cup, samples are boiled, denatured, and reduced to elute subunits for streptavidin blotting. Streptavidin, conjugated Alexa Fluor™ 647 positive bands denote reversibly oxidised subunits. (**B**). A predicted reversibly oxidised subunit profile based on Table 1. (**C**). Representative image of an experimentally observed reversibly oxidised subunit profile alongside a molecular weight (MW) ladder. Arrows indicate the predicted identity of the observed bands. The image shows several F_1_-F_o_ ATP synthase subunits are reversibly oxidised. An unpredicted band at 100 kDa was observed (see main text). Clickable TCO-Tz immunocapture coupled to redox affinity blotting was performed on five pools of 10 *X. laevis* oocytes. Each lane represents the weighted mean of 10 oocytes.

**Figure 4 antioxidants-09-00215-f004:**
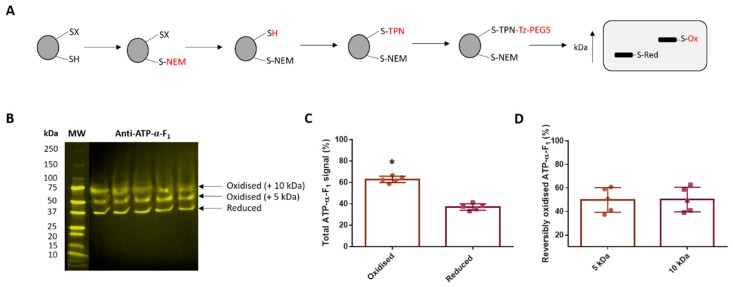
Reversible ATP-α-F1 oxidation is significant in oocytes. (**A**). Catalyst-free trans-cyclooctene-6methyltetrazine (TCO-Tz) Click PEGylation scheme for reversibly oxidised thiols. Left to right: Reduced thiols are alkylated with NEM. Reversibly oxidised thiols are reduced with TCEP before being alkylated with TCO-PEG3-NEM (TPN). TPN labelled thiols are incubated with Tz-PEG5 to initiate the catalyst-free IEDDA Click reaction. Reversibly oxidised thiols are then mass shifted when assessed by Western Blot owing to a PEG induced electrophoretic mobility shift. (**B**). Western blot image showing reversibly oxidised (i.e., mass shifted 5 and 10 kDa bands) relative to reduced ATP-α-F_1_ (i.e., lower band) in *X. laevis* oocytes (*n* = 5). MW = molecular weight. (**C**). Percent reversibly oxidised (i.e., mass shifted) compared to reduced (unshifted) ATP-α-F_1_ quantified. Percent reversibly oxidised ATP-α-F_1_ is significantly (*p* = 0.0007) greater than the amount of reduced ATP-α-F_1_. An asterix denotes statistical significance as assessed by a paired Student’s t-test. (**D**). Quantified percentage contribution of the 5 and 10 kDa bands to the total mass shifted (i.e., reversibly oxidised) signal. No significant difference (*p* = 0.09611) in the contribution of the 5 and 10 kDa band signal was observed as assessed by a paired Student’s t-test. Each *n* is the weighted mean of 10 oocytes.

**Figure 5 antioxidants-09-00215-f005:**
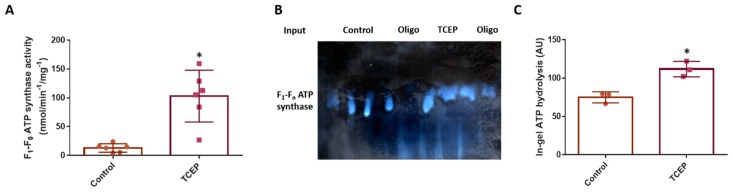
Reversible thiol oxidation inhibits the F_1_-F_o_ ATP synthase. (**A**). Chemically reversing thiol oxidation using TCEP substantially increases F_1_-F_o_ ATP synthase activity. Specifically, oligomycin sensitive F_1_-F_o_ ATP synthase activity is significantly greater (*p* = 0.0007) in TCEP (*n* = 6) compared to control oocytes (*n* = 6) in *X. laevis*. (**B**). Inverted Hr-CNPAGE image of F_1_-F_o_ ATP synthase mediated in-gel ATP hydrolysis. No signal is observed in oligomycin treated controls and a decreased signal in the TCEP condition. (**C**). Densitometry based quantification reveals a significant (*p* = 0.0067) increased F_1_-F_o_ ATP synthase mediated ATP hydrolysis in TCEP (*n* = 3) compared to control oocytes (*n* = 3) in *X. laevis*. Statistical significance is indicated by an asterix as assed by an independent Student’s t-test. Each *n* is the weighted mean of 10 oocytes.

**Table 1 antioxidants-09-00215-t001:** Cysteine residues in F_1_-F_o_ ATP synthase in *X. laevis*. No annotated information could be found for ATP synthase subunit epsilon (ATP-ε-F_1_), subunit DAPIT, ATP synthase F(0) complex subunit C2, and ATP synthase subunit e (subunit e).

ATP Synthase Subunit	Uniprot ID	Domain	Molecular Weight (kDa)	Cysteine Residues
Subunit a	P00849	F_o_	25	None
Subunit ACL	P03931	F_o_	6.5	None
C domain-containing protein (subunit c)	A0A1L8HIH0	F_o_	16.9	34 *, 49 *, 84, 149
Coupling factor 6	Q6PG55	F_o_	12.3	None
Subunit C3 (subunit c3)	Q8AVE1	F_o_	14.7	4 *, 131
Subunit f	A0A1L8EX92	F_o_	10.4	None
Subunit g (subunit g)	Q66L24	F_o_	11	96
Subunit alpha (ATP-α-F_1_)	Q68EY5	F_1_	60	244, 294
Subunit beta (ATP-β-F_1_)	A0A1L8HHY6	F_1_	56.4	9 *@, 20 *, 31 *
Subunit gamma (ATP-γ-F_1_)	Q6INB6	F_1_	32.4	100, 173 #
Subunit delta (ATP-δ-F_1_)	Q66KY9	F_1_	16.9	None
Oligomycin sensitivity conferring protein (OSCP)	Q3KQC0	F_1_	22.8	139
Subunit b (subunit b)	Q9IAJ7	F_1_	28.2	26 *, 104, 235

* Likely in the mitochondrial leader sequence; @ Only present on the L chromosome copy of the protein; # Only present on the S chromosome copy of the protein.

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
