# Peer review of "Reversible Thiol Oxidation Inhibits the Mitochondrial ATP Synthase in *Xenopus laevis* Oocytes"

_antioxidants, 2020, doi:10.3390/antiox9030215_

Round 1

Reviewer 1 Report

Here, the authors present evidence that ATP synthase undergoes S-thiolation in oocytes.

The authors refer to “oxidative phosphorylation” as “chemiosmotic oxidative phosphorylation” throughout the manuscript, as if to imply there are several types of oxidative phosphorylation pathways. Please correct to just “oxidative phosphorylation”. That said, the introduction requires a lot of revision. It is disjoined and lacks flow. The authors also make statements in the first paragraph that are not consistent with the current literature (e.g. H2O2 reacts with DNA and that copper and iron are bound to DNA).

I am unclear on the relationship between inhibition of ATP synthase, quiescence, and inhibition of ROS production. In fact, deactivation of ATP synthase is well-documented to augment ROS production, which can be easily replicated using isolated mitochondria treated with oligomycin, which increases ROS production by hyper-polarizing the MIM. The authors should measure ROS production to draw links between ATP synthase S-thiolation and the statement that oocyte quiescence is required to curtail ROS production for cell preservation.

The results collected do have merit. However, it would also be worthwhile if the authors measured the different states of respiration and calculate the respiratory control ratio, a proxy measure for the efficiency of oxidative phosphorylation.

 I am more inclined to conclude that S-thiolation of ATP  synthase is to protect it from irreversible oxidative deactivation, much like complex I and alpha-ketoglutarate dehydrogenase.

Author Response

Reviewer 1

Here, the authors present evidence that ATP synthase undergoes S-thiolation in oocytes. The authors refer to “oxidative phosphorylation” as “chemiosmotic oxidative phosphorylation” throughout the manuscript, as if to imply there are several types of oxidative phosphorylation pathways. Please correct to just “oxidative phosphorylation”.

We thank reviewer one for highlighting how our use of the term chemiosmotic oxidative phosphorylation could be interpreted to mean several types of oxidative phosphorylation mechanisms exist. In the revised manuscript, we have used the term oxidative phosphorylation (OXPHOS) throughout.

That said, the introduction requires a lot of revision. It is disjoined and lacks flow.

We agree. To improve flow, we have substantially revised the original introduction. To do so, we have: (1) added/reworded the opening sentence of each paragraph to link it to the last sentence of the previous paragraph; (2) added an additional paragraph to properly introduce each topic.

The authors also make statements in the first paragraph that are not consistent with the current literature (e.g. H2O2 reacts with DNA and that copper and iron are bound to DNA).

Perhaps, a sentence has been misread. The high activation energy associated with breaking the O-O bond prevents H2O2 from reacting with DNA bases directly. However, it is possible for H2O2 to react with DNA bound transition metals to produce hydroxyl radical. Work from the mid 1990’s evidences the feasibility of the mechanism:  https://www.ncbi.nlm.nih.gov/pubmed/7615572 and https://www.ncbi.nlm.nih.gov/pubmed/9192816 . The diffusion-controlled reactivity of hydroxyl radical means it must be produced close to the source. To clarify, we have revised the sentence to read:

“Superoxide anion and its dismutation product hydrogen peroxide (H2O2) are chemically unable to oxidise DNA directly [8]. H2O2 can, however, react with DNA bound iron and copper ions to produce hydroxyl radical (.OH) [9–11]. In turn, .OH can damage pyrimidine and purine bases at a diffusion controlled rate (i.e., k ~ 109 M-1 s-1) via addition, oxidation, and abstraction reactions [12–14].”

I am unclear on the relationship between inhibition of ATP synthase, quiescence, and inhibition of ROS production. In fact, deactivation of ATP synthase is well-documented to augment ROS production, which can be easily replicated using isolated mitochondria treated with oligomycin, which increases ROS production by hyper-polarizing the MIM. The authors should measure ROS production to draw links between ATP synthase S-thiolation and the statement that oocyte quiescence is required to curtail ROS production for cell preservation.

The reviewer makes an excellent point. We have added a paragraph explaining how dual inhibition of the proton pumps and synthase would be required to curtail mitochondrial superoxide production without sacrificing oocyte viability. The extract is:

“Allen [15] posits that: oocytes safeguard mtDNA homoplasmy by repressing OXPHOS to curtail superoxide production. In support, OXPHOS is repressed in oocytes compared to sperm in diverse phyla from jellyfish to mice [16–20]. Repressed OXPHOS is associated with lower mitochondrial free radical production in oocytes compared to sperm [16,17]. To curtail superoxide production by repressing OXPHOS without sacrificing oocyte viability, dual inhibition of the proton pumps (i.e., complex I, III and IV) and F1-Fo ATP synthase may be required. If the proton pumps are active and the F1-Fo ATP synthase is inactive, then a large electrochemical proton motive force (Δp) could substantially enhance superoxide production (e.g., by complex I catalysed reverse electron transfer [21]). Reciprocally, if the proton pumps are inactive and F1-Fo ATP synthase is active, then it would curtail complex I and III catalysed superoxide production, but the synthase may compromise oocyte viability by hydrolysing ATP to maintain Δp [22]. If and how the proton pumps and the F1-Fo ATP synthase are inhibited in oocytes is unknown. Unravelling if and how the F1-Fo ATP synthase is inhibited would, therefore, advance current understanding of reproductive biology”.

Measures of ROS production will be reported in a different paper. Based on existing data, we have added that mitochondrial ROS production is comparatively low in X. laevis oocytes. Additionally, to support parallel inhibition of the proton pumps, we have referenced a report documenting repressed cytochrome c oxidase activity in X. laevis oocytes in the revised discussion:

“cytochrome c oxidase (i.e., complex IV) activity is repressed and H2O2 levels are low in X. laevis oocytes [23,46]

The results collected do have merit. However, it would also be worthwhile if the authors measured the different states of respiration and calculate the respiratory control ratio, a proxy measure for the efficiency of oxidative phosphorylation.

We are pleased the reviewer believes our findings have merit. Respiratory analysis forms part of an on-going study because it necessitates studying the proton pumps. The advantage of the assays selected is that we could isolate the specific contribution of the synthase, which is significant when other complexes may be inhibited to curtail superoxide production.

I am more inclined to conclude that S-thiolation of ATP synthase is to protect it from irreversible oxidative deactivation, much like complex I and alpha-ketoglutarate dehydrogenase.

We have added that reversibly thiol oxidation may prevent over-oxidation to sulfinic and sulfonic acids in the discussion:

Reversible thiol oxidation may also protect the F1-Fo ATP synthase from irreversible inactivation secondary to sulfinic and sulfonic acid formation [25]

Reviewer 2 Report

This paper is interesting and opens new perspectives in reproductive biology. The F-ATPase has a recognized role in sperm function IN MAMMALS (PMID 31927418). However, author should take into account that the thiol-driven regulation of the mitochondrial F-ATPase have been previously investigated in other organisms. Interestingly, reversible oxidations may decrease the oligomycin sensitivity of the enzyme. There is also a book chapter on this topic. Redox regulation of the enzyme activity also occurs in bacterial and chloroplast ATPases, as stated in the introduction.  Protozoan an metazoan form different dimers (Front Physiol. 2018, 9:1243). The supramolecular arrangement of F-ATPase in Xenopus should be considered. Perhaps present data may be re-evaluated on the basis of these reports.

The complete scientific name (Latin) of the species should be provided in the title and in the abstract. Introduction: Hydroxyl radical should have a dot.

Lines 41,42. Why may and only are in italics?

Line 120: are

Lines 199-200. Two subsequent sentences state the same concept

What do authors mean by cognate subunits? Do they associate naturally? do they belong to the same origin?

Line 207: higher is missing

Line 225 mitochondrial (adj)

Lines 251-261 Subunits should be listed in a homogeneous way

Line 317: excluded (simple past)

The companion paper in Redox Biology vol. 26 (2019) should be cited.

The experimental design is good but data presentation and discussion can be much improved. What about the enzyme dimerization? Xenopus oocytes also contain an ectopic ATPase? Was this contribution ruled out?

There is some overlapping between Results and Discussion. Repetitions should be avoided throughout the text.

Author Response

Reviewer 2

This paper is interesting and opens new perspectives in reproductive biology.

We are pleased that the reviewer finds our work interesting and our encouraged by the new perspective in reproductive biology statement. Thank you.

The F-ATPase has a recognized role in sperm function IN MAMMALS (PMID 31927418).

The role of the mitochondrial ATP synthase in sperm function is now acknowledged in the first sentence of the introduction, with the mammalian reference added:

“Human sperm rely on oxidative phosphorylation (OXPHOS) to swim 103 times their own length to fertilise an oocyte [1,2]”.

However, author should take into account that the thiol-driven regulation of the mitochondrial F-ATPase have been previously investigated in other organisms.

While we made some attempt to cite prior work, the reference list was incomplete. Accordingly, we have cited many of the key studies that informed our hypothesis that the synthase was redox regulated in oocytes.

Interestingly, reversible oxidations may decrease the oligomycin sensitivity of the enzyme.

We agree. When an enzyme is already inhibited by reversible thiol oxidation, it is likely to be insensitive to a chemical inhibitor.

There is also a book chapter on this topic. Redox regulation of the enzyme activity also occurs in bacterial and chloroplast ATPases, as stated in the introduction.  Protozoan an metazoan form different dimers (Front Physiol. 2018, 9:1243). The supramolecular arrangement of F-ATPase in Xenopus should be considered. Perhaps present data may be re-evaluated on the basis of these reports.

The reference in Frontiers in Physiology has been added. The points raised here are addressed below.

The complete scientific name (Latin) of the species should be provided in the title and in the abstract.

We agree and the manuscript has been revised accordingly.

Introduction: Hydroxyl radical should have a dot.

Is has a dot already—to make it clearer we have increased the font size.

Lines 41,42. Why may and only are in italics?

The italics have been removed.

Line 120: are

This has been amended.

Lines 199-200. Two subsequent sentences state the same concept

What do authors mean by cognate subunits? Do they associate naturally? do they belong to the same origin?

We apologize for the confusion and have corrected this to F1-Fo ATP synthase subunits throughout.

Line 207: higher is missing

Higher has been added.

Line 225 mitochondrial (adj)

Changed.

Lines 251-261 Subunits should be listed in a homogeneous way

We agree and the gene names are now provided in the table.

Line 317: excluded (simple past)

Changed.

The companion paper in Redox Biology vol. 26 (2019) should be cited.

We are pleased that reviewer 2 is aware of our work in Redox Biology and have cited the paper in the revised manuscript.

The experimental design is good but data presentation and discussion can be much improved. What about the enzyme dimerization? Xenopus oocytes also contain an ectopic ATPase? Was this contribution ruled out?

The data presentation may relate to low quality figures inserted into the manuscript. The resubmitted version contains highly resolved images. We assume the reviewer refers to non-mitochondrial ATPase when using ectopic. If so, then yes on two accounts: (1) by isolating mitochondria and membranes; (2) by using a selective F1-Fo ATP synthase inhibitor.

Regarding the dimer point and discussion. We have extensively revised the discussion to improve it. Specifically, we have added a paragraph addressing dimers:

“From a structural perspective, S-glutathionylation could lock the F1-Fo ATP synthase in a monomeric state by impeding the formation of dimers, especially if they involve intermolecular disulfide bonds [31]. Alternatively, a negative charge may electrostatically repel dimer interfaces. Thiols in subunit e and g also regulate the stability and formation of oligomers, which dictate inner mitochondrial membrane topology [71–73]. For example, F1-Fo ATP synthase dimers shape cristae to ensure optimal ATP synthesis [74]. Perhaps, the redox state of Fo underlies the lack of mature cristae in oocytes [19]. Intriguingly, Teixeira and colleagues [75] found that OXPHOS is dispensable for germline stem cell differentiation but F1-Fo ATP synthase dimers are essential because they are required for cristae maturation. The results of their study and the present work raise the possibility of at least two redox switches: (1) to prohibit dimers; and (2) to inhibit catalysis. Two redox switches would endow mitochondria with the capacity to form mature cristae without a catalytically active F1-Fo ATP synthase. That is, to uncouple cristae maturation from OXPHOS [75]

There is some overlapping between Results and Discussion. Repetitions should be avoided throughout the text.

We agree so we have eliminated repetitive elements. For example, the repetitive concluding statements at the end of each paragraph in the results have been omitted.

Round 2

Reviewer 1 Report

None

Author Response

We thank the reviewer for their kind time and consideration.

Reviewer 2 Report

The manuscript is significantly improved. However not all the points raised in my previous report are satisfied. For instance,  I disagree with the implicit idea that changes in thiol redox state of the enzyme obligatory lead to enzyme inhibition, as shown for oocyte mitochondrial F1FO-ATP synthase. Thiol oxidation may change the protein conformation and, as shown for other enzyme complexes, may also favour catalysis. The authors' response regarding the loss of oligomycin insensitivity due to thiol oxidation as elsewhere reported, is not convincing. The decrease in oligomycin sensitivity caused by thiol oxidation may even lead to an incorrect evaluation of the enzyme activity. This possibility is not so remote and it should be at least considered (or even criticized) in the Discussion. Moreover, as far as I am aware, the maintenance of oligomycin sensitivity assesses that F1 and FO are functionally coupled and not exactly that F1 and FO remain assembled (line 313). Minor comments: lines 233 and asterisk (lowercase). Remove cognate (line 223) or explain what is meant. Line 247: OSCO or OSCP? Lines 371-372: cristae should be in italics.

Table 1: beta (lower case). subunit b (lower case). The repetition of ATP synthase in each row is unnecessary. In the heading: ATP synthase subunits

Author Response

Response to reviewer 2

The manuscript is significantly improved.

We thank the reviewer for their help in significantly improving the manuscript.

However not all the points raised in my previous report are satisfied. For instance,  I disagree with the implicit idea that changes in thiol redox state of the enzyme obligatory lead to enzyme inhibition, as shown for oocyte mitochondrial F1FO-ATP synthase. Thiol oxidation may change the protein conformation and, as shown for other enzyme complexes, may also favour catalysis.

We agree it is unwise to always assume thiol oxidation is inhibitory, especially when a redox code may exist wherein the biological outcome varies according to the identity (number of thiols modified), occupancy (relative oxidation), type (modification identity). Paragraph 3 of the discussion has been revised accordingly:

“Given reversible thiol oxidation can activate several enzymes (see [36]), it is unwise to assume reversible thiol oxidation is always inhibitory. In this regard, a redox code may exist wherein the biological outcome varies according to the number of thiols and subunits modified, occupancy (relative oxidation), and modification type [65]

The authors' response regarding the loss of oligomycin insensitivity due to thiol oxidation as elsewhere reported, is not convincing. The decrease in oligomycin sensitivity caused by thiol oxidation may even lead to an incorrect evaluation of the enzyme activity. This possibility is not so remote and it should be at least considered (or even criticized) in the Discussion.

In light of the reviewer’s point, we have amended paragraph 3 of the discussion:

“Intriguingly, reversible thiol oxidation may lead to oligomycin sensitive enzyme activity being underreported, if it impacts inhibitor binding and/or action. Follow-up studies are required to address this possibility”

Moreover, as far as I am aware, the maintenance of oligomycin sensitivity assesses that F1 and FO are functionally coupled and not exactly that F1 and FO remain assembled (line 313).

With regards to line 313, we agree with the reviewer and have elected to omit the two sentences regarding oligomycin and assembly in the results section.

Minor comments: lines 233 and asterisk (lowercase). Remove cognate (line 223) or explain what is meant. Line 247: OSCO or OSCP? Lines 371-372: cristae should be in italics.

We apologise for the above-mentioned errors and have corrected each one in the revised manuscript (see yellow highlighted sections).

Table 1: beta (lower case). subunit b (lower case). The repetition of ATP synthase in each row is unnecessary. In the heading: ATP synthase subunits

We apologise for the above-mentioned errors and have corrected each one in the revised Table.